# All-atom simulation of the HET-s prion replication

Luca Terruzzi[1,2☯], Giovanni Spagnolli[2,3☯]*, Alberto Boldrini[1,2], Jesús R. Requena[4], Emiliano Biasini[2,3]*, Pietro Faccioli[5,6]*

1 Sibylla Biotech SRL, Verona, Italy, 2 Department of Cellular, Computational and Integrative Biology (CIBIO), University of Trento, Povo, Trento, Italy, 3 Dulbecco Telethon Institute, University of Trento, Povo, Trento, Italy, 4 CIMUS Biomedical Research Institute & Department of Medical Sciences, University of Santiago de Compostela-IDIS, Spain, 5 Department of Physics, University of Trento, Povo, Trento, Italy, 6 INFN-TIFPA, University of Trento, Povo, Trento, Italy

☯ These authors contributed equally to this work.
* giovanni.spagnolli@unitn.it (GS); emiliano.biasini@unitn.it (EB); pietro.faccioli@unitn.it (PF)

**Data Availability Statement:** All relevant data are within the manuscript and its Supporting Information files.

**Funding:** This work was partially supported by a grant from Fondazione Telethon (Italy, TCP14009).

## Abstract

Prions are self-replicative protein particles lacking nucleic acids. Originally discovered for causing infectious neurodegenerative disorders, they have also been found to play several physiological roles in a variety of species. Functional and pathogenic prions share a common mechanism of replication, characterized by the ability of an amyloid conformer to propagate by inducing the conversion of its physiological, soluble counterpart. Since time-resolved biophysical experiments are currently unable to provide full reconstruction of the physico-chemical mechanisms responsible for prion replication, one must rely on computer simulations. In this work, we show that a recently developed algorithm called Self-Consistent Path Sampling (SCPS) overcomes the computational limitations of plain MD and provides a viable tool to investigate prion replication processes using state-of-the-art all-atom force fields in explicit solvent. First, we validate the reliability of SCPS simulations by characterizing the folding of a class of small proteins and comparing against the results of plain MD simulations. Next, we use SCPS to investigate the replication of the prion forming domain of HET-s, a physiological fungal prion for which high-resolution structural data are available. Our atomistic reconstruction shows remarkable similarities with a previously reported mechanism of mammalian PrP^Sc propagation obtained using a simpler and more approximate path sampling algorithm. Together, these results suggest that the propagation of prions generated by evolutionary distant proteins may share common features. In particular, in both these cases, prions propagate their conformation through a very similar templating mechanism.

## Author summary

Prions are proteins capable of replicating in absence of nucleic acids. By propagating the information encoded in their conformation, prions exemplify the phenomenon of protein-based inheritance. These peculiar agents are associated with neurodegenerative

Giovanni Spagnolli is a recipient of a fellowship from Fondazione Telethon. Emiliano Biasini is an Assistant Telethon Scientist at the Dulbecco Telethon Institute (Fondazione Telethon, Italy). The funders had no role in study design, data collection and analysis, decision to publish, or preparation of the manuscript.

**Competing interests:** Luca Terruzzi and Alberto Boldrini have direct involvement in the ongoing research at Sibylla Biotech SRL (www.sibyllabiotech.it). Giovanni Spagnolli, Emiliano Biasini and Pietro Faccioli are co-founders and shareholders of the company.

pathologies in mammals, but also involved in a wide variety of physiological processes occurring in various biological contexts along the evolutionary scale. In this work, we apply a recently developed computational method to study the propagation mechanism of the fungal prion HET-s, using a realistic all-atom model. We find that the replication of HET-s shares fundamental features with the templated conversion of the mammalian prion PrP$^{Sc}$.

## Introduction

The phenomenon of protein-based inheritance characterizes prions, proteins appearing at various levels along the evolutionary scale that are capable of propagating their conformationally encoded information in absence of nucleic acids [1]. Despite their original identification as causative agents of neurodegenerative conditions in mammals, prions also exert regulatory roles in different biological contexts [2, 3]. For example, a mechanism of heterokaryon incompatibility in different fungi is regulated by a prion [3–5]. This process reflects the inability of vegetative fungal cells from two different strains to undergo fusion, depending on specific *loci* (het) whose alleles must be identical for stable hyphal fusion to occur. Strain compatibility ultimately determines whether the heterokaryon develops normally or undergoes controlled cell-death. In *Podospora anserina*, the heterokaryon incompatibility is specified by a het *locus* appearing as two distinct and incompatible alleles (*HET-s* and *HET-S*), encoding two corresponding proteins (HET-s and HET-S, respectively) [6]. When a HET-s strain fuses with another expressing HET-S, the heterokaryon can undergo controlled cell death. However, incompatibility occurs only when the HET-s factor is folded in an amyloid prion conformation. This trait can be inherited over many generations due to the ability of the HET-s prion to catalyze the conversion of soluble HET-s molecules into the growing amyloid fibrils [7].

The 2-rung-β-solenoid (2RβS) architecture of the HET-s prion has been solved at high resolution by solid-state NMR [8]. On the other hand, the elucidation of the mechanism underlying its propagation is still missing, mainly due to the lack of suitable biophysical methods to characterize such molecular processes at high spatiotemporal resolution.

Computational techniques, such as MD, may in principle be employed to achieve the required level of resolution. However, plain MD simulations can only be applied to study large conformational transitions occurring up to the microsecond timescales [9]. In contrast, folding and misfolding of most proteins and large conformational re-arrangements including prion replication occur at much longer timescales, ranging from milliseconds to minutes [10].

Using enhanced sampling algorithms, it is possible to explore protein conformational states much more efficiently than by plain MD (for recent reviews of some of these methods see e.g. [11, 12]), at the price of having to introduce additional approximations or supply prior information. In particular, in the so-called ratchet-and-pawl MD (rMD) [13, 14], an unphysical history-dependent biasing force is introduced to prevent the chain from backtracking along the direction defined by an arbitrarily chosen Collective Variable (CV). Conversely, the biasing force remains latent when the system spontaneously progresses towards the product. It has been rigorously shown that rMD samples the Boltzmann distribution in the transition region when the CV used is the committor function [15]. In practical calculations, the chosen CV can at most represent a reasonable proxy of the committor. In this case, it is still possible to keep the systematic errors of rMD simulations to a minimum by applying the so-called Bias Functional (BF) method [16]. In this approach, a variational principle derived from the Langevin

dynamics is applied to score the reactive trajectories generated by rMD, in order to identify those with the highest probability to occur in the absence of any biasing force.

Protein folding pathways obtained with the BF approach have been found to agree very well with the results of both plain MD simulations [16] and kinetic experiments [17, 18], arguably reflecting the fact that a good CV for protein folding is available [19], as suggested by energy landscape theory arguments [20]. Unfortunately, the BF approach may be flawed by uncontrolled systematic errors whenever it is applied to study processes in which the reaction coordinate is poorly known.

In our previous work, we used rMD simulations to propose a first fully atomistic model for the replication of the mammalian PrP$^{Sc}$ prion [21]. To perform such a calculation, first, we obtained an estimate of the reaction coordinate using a phenomenological stochastic model. Then, we used this CV in the definition of the biasing force. The resulting prion conversion mechanism was one in which PrP$^C$ sequentially unfolded and progressively bond to the prion fibril, through a rung-by-rung templated mechanism. A key limitation of this approach is that the results of the simulation significantly depend on the model used to define the CV. Using the BF protocol one may hope to reduce possible systematic errors. Yet, it is reasonable to expect that even the BF results to be strongly model dependent.

In this work, we show that this problem is overcome by the recently developed SCPS algorithm [22], described in Methods. Like the BF approach, SCPS is powered by rMD-type simulations. However, unlike in BF simulation, the reaction coordinate of SCPS is not heuristically postulated. Instead, it is calculated self-consistently, through an iterative process (see Fig 1). It has been shown that SCPS provides a rigorous mean-field approximation of the unbiased Langevin dynamics, even when the initial guess of reaction coordinate is suboptimal [15, 22]. Thus, SCPS provides a much better tool than BF to investigate structural reactions in which the RC is poorly known, including prion propagation. The computational cost of SCPS, however, is about one order of magnitude larger.

To date, the SCPS scheme was only tested in a single proof-of-principle simulation of the folding of a 35 amino-acid long protein subdomain, using an implicit solvation model [22]. As a preliminary step, we performed a more extensive validation, by studying the folding of set 5 different proteins which differ in size, topology and secondary structure composition using a state-of-the-art force field in explicit solvent. Remarkably, the folding pathways of all these chains were found to be statistically indistinguishable from those obtained in the same force field using plain MD simulations on the Anton supercomputer [23].

Next, we used the SCPS scheme to simulate the misfolding of the HET-s prion forming domain and incorporation into a growing fibril. We found that the all-atom reconstruction of the HET-s replication mechanism is characterized by the templated, progressive folding of an unstructured HET-s monomer onto the exposed edges of the growing prion fibrils. Thus, on the qualitative level, these results fully confirm those previously obtained by rMD, in our study of mammalian prion replication.

## Results/Discussion

### SCPS validation in explicit solvent

To assess the accuracy of the SCPS approach we report a benchmark based on 5 different proteins, whose folding pathways have been fully characterized by plain MD [23]: (i) Trp-Cage; (ii) Villin headpiece; (iii) WW domain; (iv) NTL9 and (v) the thermostable variant of the Engrailed homeodomain. This set includes members of the three major structural classes (α-helical, β-sheet and mixed α/β).

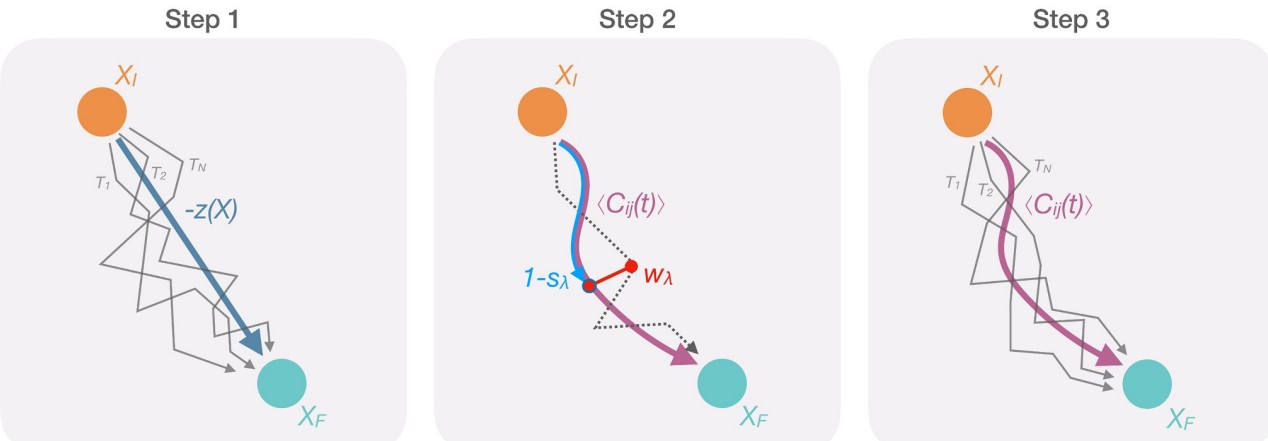

**Fig 1. Schematic representation of the SCPS algorithm.** $X_I$ represents the initial protein conformation, $X_F$ the final target conformation and $T_{1...N}$ the trajectories connecting $X_I$ to $X_F$. **Step 1**: an ensemble of trajectories starting from $X_I$ and successfully reaching $X_F$ is generated by employing the rMD, biasing along the pre-defined CV $z(X)$. Since $z(X)$ decreases when the proximity of $X$ to $X_F$ increases, the progression along the path is here defined as $-z(X)$. **Step 2**: the trajectories successfully reaching the target state are then used to compute the mean path $\langle C_{ij}(t) \rangle$, depicted in purple (see Methods). The mean path is then used to define two new coordinates: $s_\lambda$, depicted in blue, which value is 1 in the unstructured state and 0 in the target state, therefore $(1 - s_\lambda)$ is used here to define the progress along the mean path; and $w_\lambda$, depicted in red, that represents the distance to the mean path. **Step 3**: a modified version of the rMD is employed to generate a new set of trajectories by introducing two biasing forces, acting along $s_\lambda(t)$, and $w_\lambda(t)$ instead of $z(X)$. The trajectories successfully reaching the target state are then used to compute a new mean path (step 2) to perform a new iteration.

To assess whether the rMD and SCPS trajectories significantly match plain MD data, we employed a similarity definition of trajectories based on the order of contact formation [14]. This metrics, $s(k, k_0)$, is equal to 1 when all the native contacts between trajectory $k$ and $k_0$ are formed in the same order, while it is 0 if they are formed in a completely different order (see Eqs 9 and 10 in Methods). First, for each protein we investigated the similarity of the folding events sampled with plain MD within themselves. This step generated a self-similarity distribution (denoted as $A$). Folding events sampled with biased MD (rMD or SCPS) were then compared to the those sampled with plain MD, generating cross-similarity distributions (denoted as $R_i$, where $i$ is the iteration number). An additional cross-similarity random distribution, $R_r$, used as a reference, was generated by comparing plain MD simulations with random sequences of native contact formation. These distributions are reported in S1 Fig in the Supplementary Information (SI).

To quantify the degree of agreement of our simulations with the results of plain MD, we performed an analysis based on the Kullback-Leibler divergence ($D_{KL}$) [24]. $D_{KL}$ is a measure of difference between two distributions $R$ an $A$, and represents the information loss when the distribution $R$ is used to approximate $A$ (see Methods). Specifically, at each SCPS iteration we computed the $D_{KL}$ between the cross-similarity distribution between SCPS and MD trajectories ($R_i$) and the self-similarity distribution of the MD folding paths $(A)$. As a reference, we also computed the $D_{KL}$ between the cross-similarity distribution $R_r$ calculated comparing random sequences of native contact formation with MD trajectories and $A$. The results reported in Fig 2 show that SCPS produces folding mechanisms that are statistically indistinguishable from those sampled by plain MD, with no significant loss of information. In 4 out of 5 of the simulated proteins, the $D_{KL}$ distance between $R_i$ and $A$ is consistently lower than the distance between the random cross-similarity $R_r$ and $A$. The only exception is Trp-cage, for which both $R_i$ and $R_r$ are indistinguishable from $A$. Such an exception can be explained by the trivial topology of the Trp-Cage and implies a very heterogeneous folding mechanism, in which many different sequences of native contact formation can be realized with comparable probability.

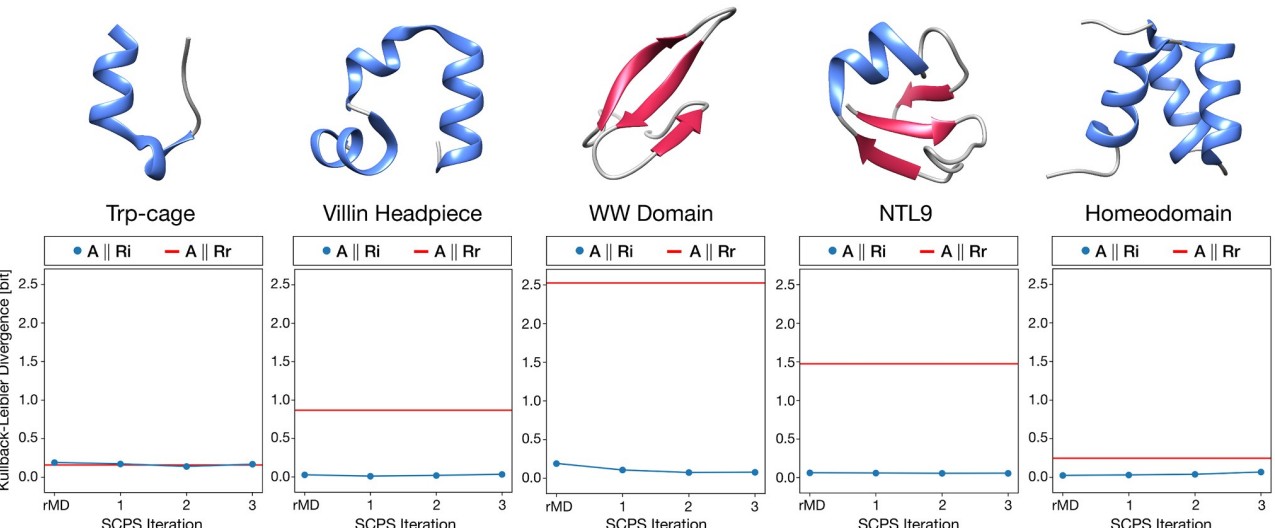

**Fig 2. Comparison between plain MD and biased (rMD and SCPS) trajectories.** In each plot, the Kullback-Leibler divergence between the cross-similarity distribution ($R_i$) of each iteration $i$ and the self-similarity distribution ($A$) is shown with blue dots. Each cross-similarity distribution $R_i$ represents the path similarity distribution between the biased and the plain MD reactive trajectories. The self-similarity distribution is the path similarity distribution within the folding events sampled with plain MD. The average Kullback-Leibler divergence between the random cross-similarity distributions ($R_r$) and the self-similarity distribution is depicted as a red line and it is used as a reference. The random cross-similarity distribution was generated by comparing plain MD simulations with randomly sampled sequences of native contact formation. These results show that both rMD and SCPS produce results that are identical to the one generated with plain MD and are different (with the exception of Trp-cage) from the randomly generated events.

We note that the application of the SCPS scheme did not lead to significant improvement over rMD, because the folding events generated by rMD are already in very good agreement with MD. This finding is consistent with the notion that the CV $z(X)$ used in rMD simulations correlates well with the fraction of native contacts, which is considered a good reaction coordinate for the folding of small globular proteins [19,20]. On the other hand, we expect SCPS and plain rMD to differ substantially where the initial guess of RC is poorer, as it is arguably the case for prion replication. This situation may be encountered also when simulating the folding of large and topologically complex proteins.

## All-atom reconstruction of HET-s prion propagation

Enhanced path sampling simulations, including SCPS, require a model for the reactant and product states. In our study of HET-s prion propagation, we used as product state the amyloid structure of the HET-s prion forming domain, which includes a trimer previously solved by solid-state NMR (PDB 2KJ3, Fig 3). The monomeric soluble state of the HET-s prion-forming domain is unstructured [25]. Thus, to generate the initial conditions in the reactant state, we performed high-temperature MD simulations on the HET-s trimer introducing positional restraints on heavy atoms on two out of the three monomers. We obtained a total of 10 initial unstructured conditions: 5 initial conditions in which the unstructured monomer resides at the N-terminus (by introducing the restraints on the two C-terminal monomers), and 5 initial conditions in which the unstructured monomer resides at the C-terminus (by introducing the restraints on the two N-terminal monomers; S2 Fig in SI). For each initial condition in the reactant state, we applied the SCPS protocol and retained only the trajectories which successfully reached the product state.

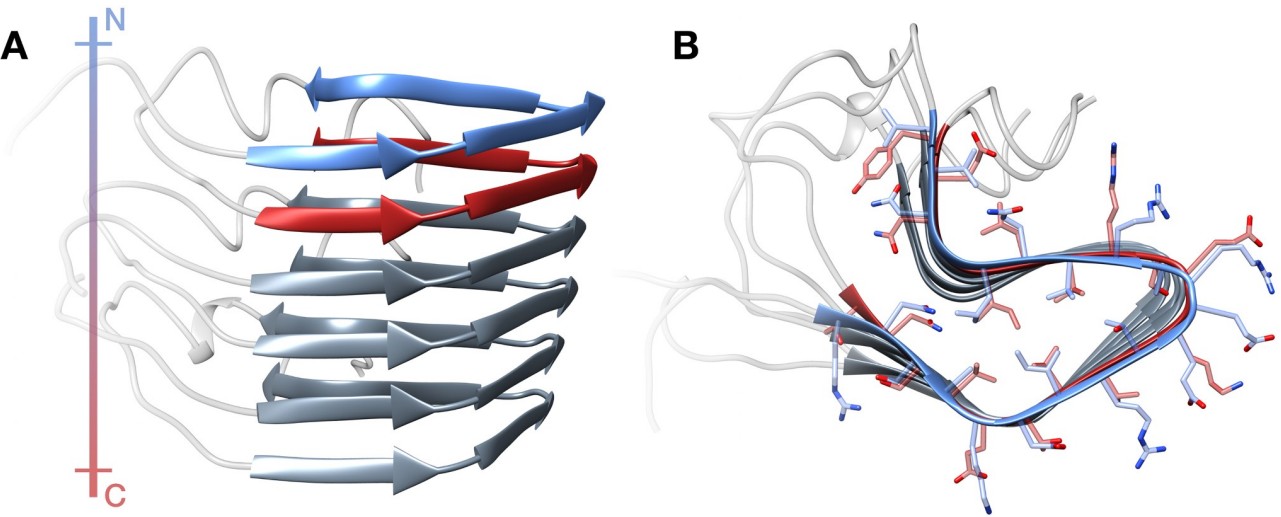

**Fig 3. Structure of the HET-s Prion Forming Domain in the amyloid form.** Lateral (**A**) and top (**B**) view of a HET-s amyloid trimer retrieved from PDB 2KJ3. Each monomer of the fibril displays a 2-Rung-β-Solenoid conformation. The N-terminal rung (residues 225–245) is depicted in blue, while the C-terminal rung (residues 261–281) is depicted in red. The colored bar at the left represents the polarity (N to C, blue to red) of the fibril.

The results obtained are reported in Fig 4, where we represent the conformations explored by the trajectories projected on the plane defined by the root mean square deviation (RMSD) of atomic positions, to the reference state of each rung of the converting monomer. Namely, the heat map displays the quantity $G(RMSD_{CT-Rung}, RMSD_{NT-Rung}) = -ln[P(RMSD_{CT-Rung}, RMSD_{NT-Rung})]$, where $P(RMSD_{CT-Rung}, RMSD_{NT-Rung})$ is the probability of observing specific RMSD pairs, calculated from a frequency histogram of the SCPS trajectories.

The entire set of productive SCPS trajectories converged on a single reaction path, which proceeds by forming the rung anchoring the fibril end, followed by the formation of the subsequent rung. These data support a picture according to which the process of HET-s prion propagation occurs by a templating mechanism (S1 and S2 Movies in SI).

To further characterize the reaction process underlying the templated conversion of HET-s prions, we performed additional analyses. In Fig 5, we show the median value of the reaction progress variable $Q$ at which each residue assumes the β-strand conformation. Residues forming at high values of $Q$ assume β-strand conformation in the late stage of the reaction. The sequence of secondary structure formation can be interpreted as the order with which monomer residues dock to the fibril. In the case of initial anchoring occurring at the N-terminus of the HET-s fibril, the C-terminus of the incoming monomer acquires a β-sheet conformation by progressively establishing intermolecular hydrogen bonds with the exposed edge residues. Once completed, the new rung templates the formation of intramolecular hydrogen bonds with the remaining residues of the polypeptide. Insertion of monomers at the C-terminal edge of the HET-s fibril proceeds in a specular fashion. These observations indicate that the two exposed edges of the HET-s amyloid provide the initial scaffold for the templated conversion of incoming monomers, which become new edges after completion of the reaction. Interestingly, while the C-terminal strand shows significant variation in the order of secondary structure formation, the N-terminal strand appears to fold in a much more stereotypical fashion. For example, residues 238–241 fold prior to 226–234, regardless of the direction of the propagation. This observation may suggest an experimentally testable hypothesis, such as that mutating the former residues could have a stronger effect on fibril formation as compared to the latter.

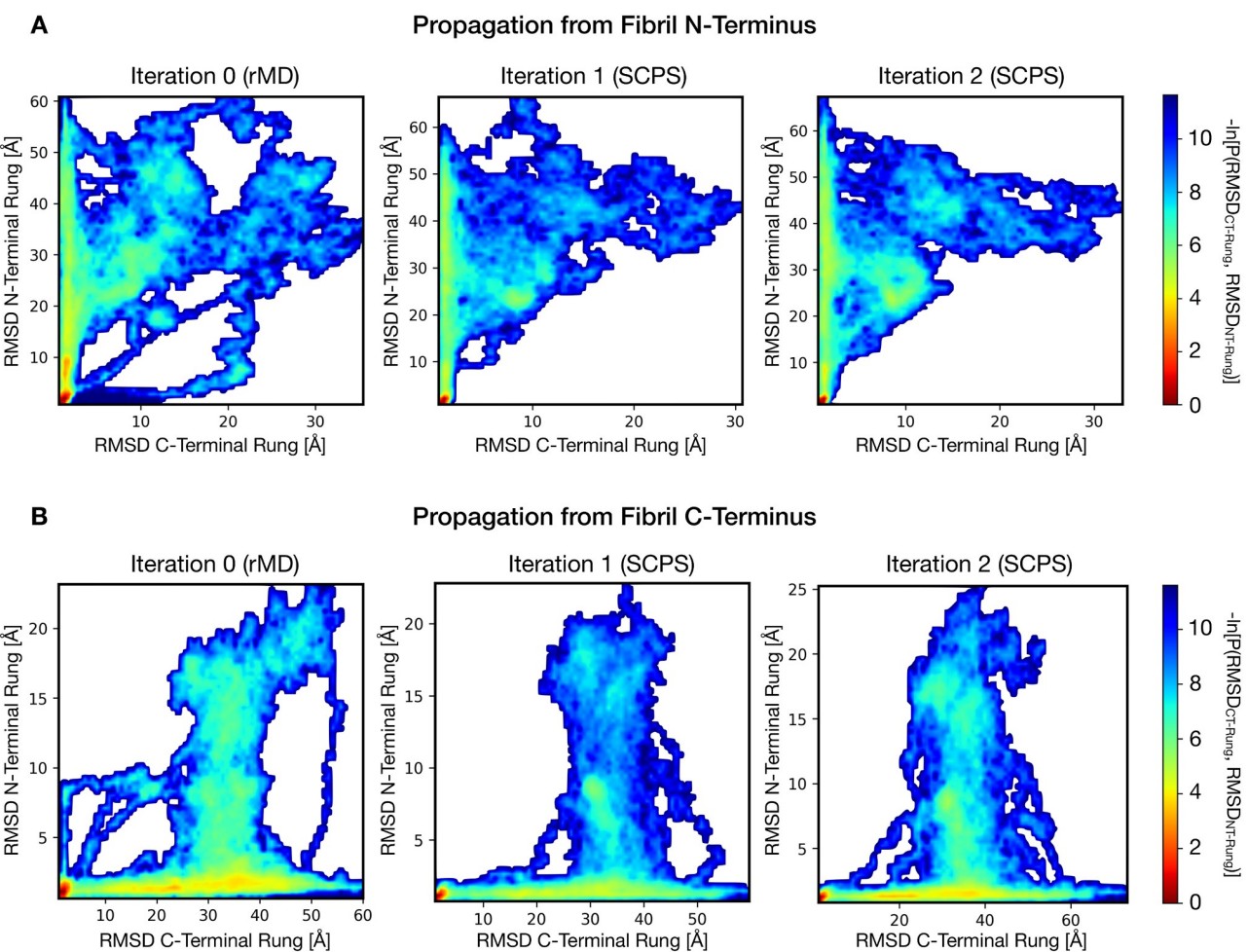

**Fig 4. Reaction pathways in HET-s rungs formation.** The heat-maps represent the negative logarithm of the probability density calculated from the SCPS reactive paths at different iterations, as a function of the RMSD from the target structure of the N- and T-terminal rungs, i.e. $-\ln[P(\text{RMSD}_{\text{CT-Rung}}, \text{RMSD}_{\text{NT-Rung}})]$. **A.** Graphs relative to trajectories propagating from the fibril N-terminus. **B.** Graphs relative to the trajectories propagating from the fibril C-terminus. In both cases, a prominent pathway, consisting of the consecutive formation of rungs starting at the fibril end, begins to appear in the rMD generated trajectories. However, the rMD algorithm also yields pathways with cooperative or inverted rung formation. These alternative events are disappeared after a single SCPS iteration. Of note, a second SCPS iteration does not produce a consistent change in the free energy landscape in both A and B, indicating the convergence of the algorithm.

To obtain an unbiased projection of the reaction paths without selecting *a priori* the collective variables for its representation, we performed a principal component analysis (PCA) on the Cα contact maps of the sampled transition pathways (Fig 6A and 6D). The PCA energy landscape representation showed that in both N-terminal and C-terminal propagation pathways the reaction initially occurs almost exclusively along the principal component 2 (PC2), and only subsequently toward the principal component 1 (PC1). Further analysis of the contribution of each contact distance to the two principal components was performed by grouping the contacts in three sets: (i) contacts between residues belonging to the same rung (intra-rung); (ii) contacts between residues of different rungs (inter-rung); (iii) contacts between residues of the converting monomer and the structured fibril (monomer-fibril) (Fig 6B and 6D). This analysis revealed that the main contact contributors for PC1 belong to the inter-rung set, while contributors of PC2 belong mainly to the monomer-fibril set. The PCA analysis repeated using an all-atom contact map yielded to overlapping results (S3 Fig in SI).

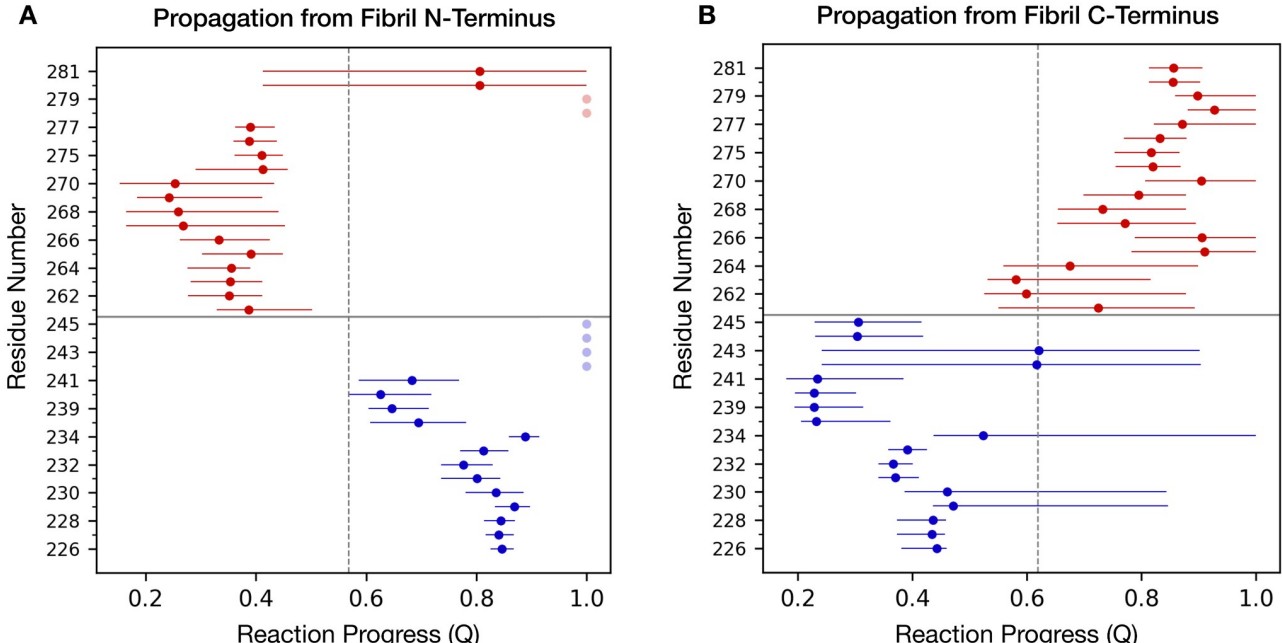

**Fig 5. Order of β-strand formation along the HET-s propagation pathway.** The median value of the reaction progress variable Q at which each residue of the rungs assumes the β-strand conformation is reported. Blue dots correspond to residues in the N-terminal rung while red dots correspond to residues in the C-terminal rung. Dots shown in transparency indicate residues not achieving a stable β-strand conformation. Horizontal bars span between the first and the third quartile of the distribution. The vertical dashed line delineates the average reaction progress at which half of the rungs-residues are incorporated into β-strand conformation. In the propagation starting from the fibril N-terminus (**A**) the residues of the C-terminal rung of the converting monomer are incorporated first (mean Q = 0.44), followed by the residues in the N-terminal rung (mean Q = 0.69). The opposite sequence of events is observed when propagation starts from the fibril C-terminus (**B**): the residues of the N-terminal rung of the converting monomer are incorporated first (mean Q = 0.42), followed by the residues in the C-terminal rung (mean Q = 0.81).

Overall, these findings corroborate a rung-by-rung propagation model for HET-s. Interestingly a highly populated region appears at the elbow of the landscape graph, in both the N-terminal and C-terminal fibril growths. Such a region reflects long-living misfolding intermediates occurring during the HET-s templated conversion mechanism, characterized by the presence of only one rung of the converting monomer attached to the elongating fibril (Fig 6C and 6F).

Collectively, our simulations indicate that HET-s prion conversion occurs through a series of templating events occurring at the N-terminal and C-terminal ends of the fibrils. In these sites, incoming HET-s monomers are initially incorporated by establishing intermolecular hydrogen bonds with the exposed β-strands, leading to the formation of a first rung, which then forms new intramolecular hydrogen bonds with the remaining parts of the polypeptide, giving rise to a new 2RβS fibril subunit (Fig 7).

We emphasize that the results of our SCPS simulation of the entire sequence of events underlying the incorporation of HET-s monomers into fibrils were obtained without relying on a specific choice of RC, i.e. the absence of strong *ad-hoc* assumptions concerning the reaction mechanism.

The stochastic model developed in Ref. [21] predicts that in the presence of a template the dominant reaction pathway should be one in which prion rungs misfold one-by-one and directly incorporate into the templating fibril, as is observed in our SCPS simulation. The statistical model introduced in Ref. [26] uses a similar theoretical argument to predict that misfolding should be highly promoted by the seed fibril. This is in line with the experimental

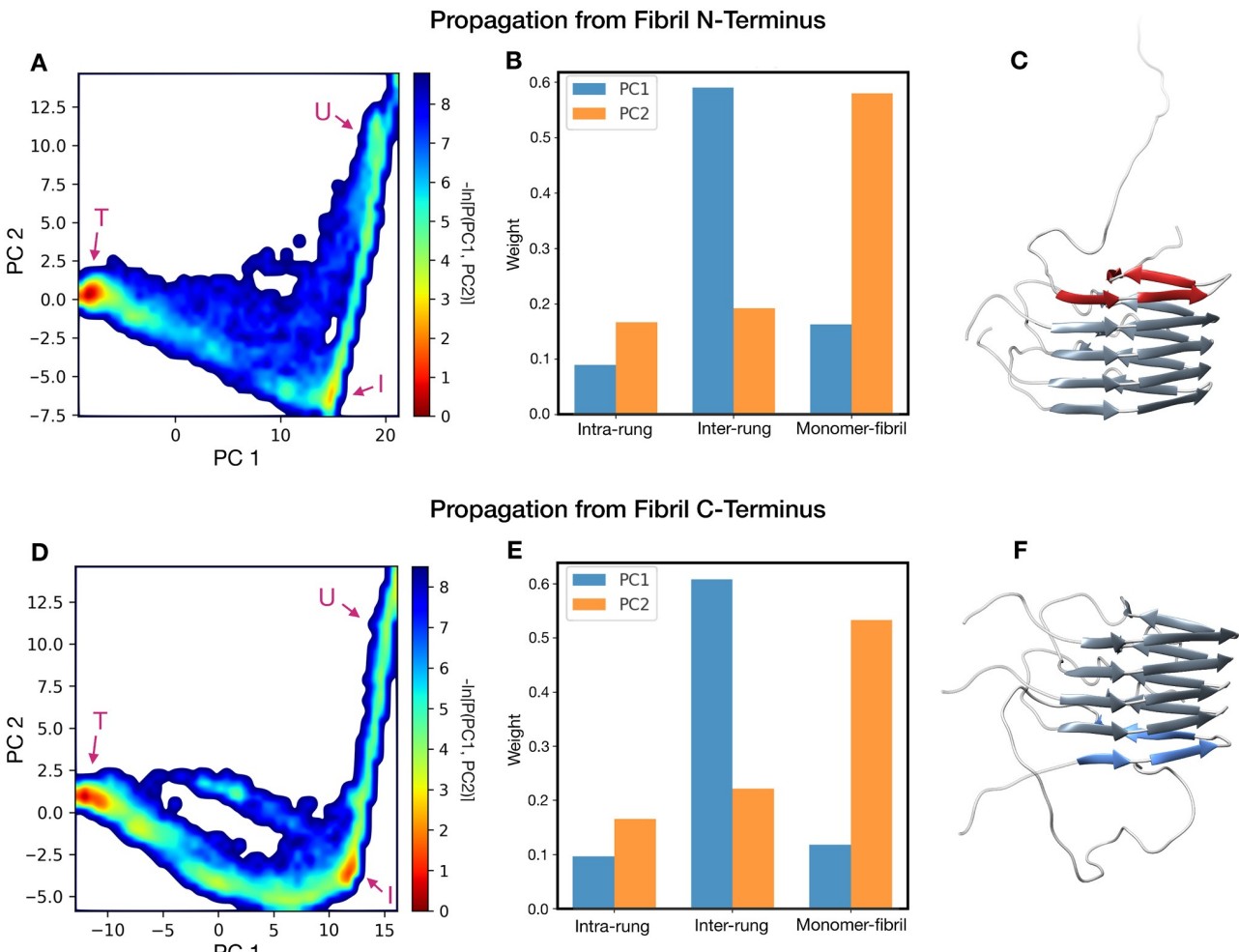

**Fig 6. Principal component analysis of the HET-s propagation trajectories.** Graphs (**A**) and (**D**) represent the free energy landscape in the principal component plane of the trajectories propagating from the fibril N- and C-terminus respectively. The letters "U", "I" and "T" indicate the unstructured, intermediate the final (target) state, respectively. The residue contacts were classified in three categories: contacts between Cα in the same rung (intra-rung), contacts between Cα belonging to different rungs (inter-rung) and contacts between Cα of the converting monomer and Cα of the structured fibril (monomer-fibril). The contribution of these sets of the two principal components is shown in bar plots (**B**) for the N-terminal propagation and (**E**) for the C-terminal propagation. Images (**C**) and (**F**) show representative protein conformations sampled from the intermediate state, for the N-terminal and C-terminal propagations, respectively.

evidence showing that in seeds-free conditions the spontaneous fibrillization of soluble HET-s conformers is characterized by a lag phase (nucleation process), which is instead abrogated in the presence of HET-s prion seeds [27].

## Comparison between the replication mechanism of PrP^Sc and HET-s prions

The main difference between the replication mechanisms of PrP$^{Sc}$ and HET-s prion in our simulations is the monodirectional growth of the former in contrast to the bidirectional extension of the latter. The reason for such difference is related to the conformation of the monomers acting as substrates. The propagation of PrP$^{Sc}$ involves the refolding of both structured (C-terminus) and unstructured (N-terminus) domains of PrP$^{C}$, while the propagation of HET-s involves exclusively the intrinsically disordered prion-forming domain of the protein.

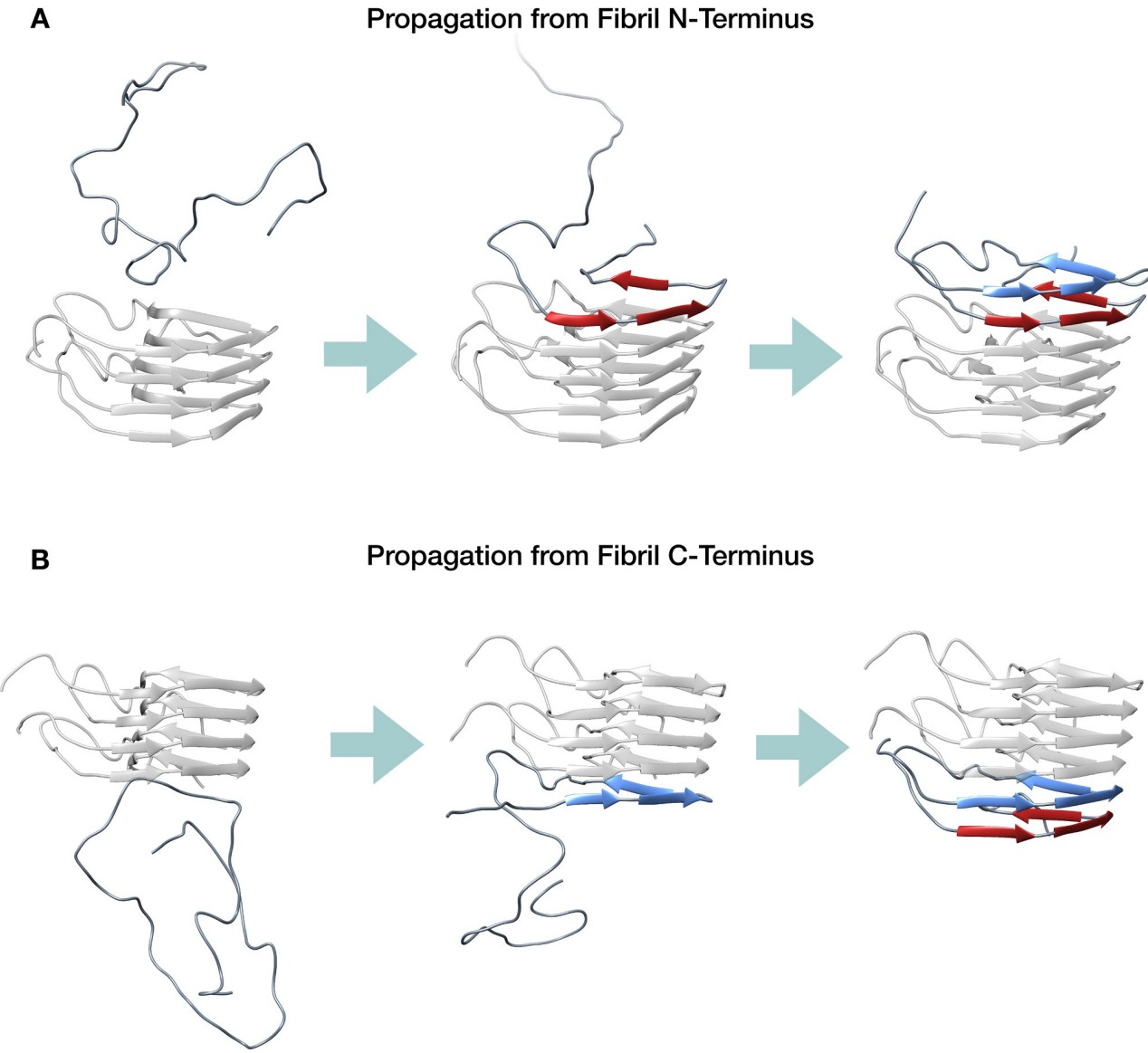

**Fig 7. Representative scheme for HET-s prion propagation. A**. Propagation scheme starting from the N-terminal end of the fibril. In this process, the C-terminal rung of the converting monomer (depicted in red) is formed by templating onto the structured N-terminal rung of the fibril (shown in transparent grey). Subsequently, the N-terminal part of the converting monomer forms the second rung (depicted in blue); **B**. Propagation scheme starting from the C-terminal end of the fibril. In this process, the N-terminal rung of the converting monomer (depicted in blue) is formed by templating onto the structured C-terminal rung of the fibril (shown in transparent grey). Subsequently, the C-terminal part of the converting monomer forms the second rung (depicted in red).

Such a difference implies that, while the replication mechanism of PrP$^{Sc}$ has an intrinsic directionality determined by the initial binding and subsequent refolding of parts of the unstructured N-terminal domain, the HET-s prion may replicate by engaging interactions with both the N-terminus or C-terminus of its substrate. Regardless of such a difference, PrP$^{Sc}$ and HET-s prions share an almost identical propagation mechanism, characterized by a progressive, rung-by-rung pathway, governed by the cooperativity of hydrogen bonds and the lateral stacking of the side chains. Another possibility is that, while our method cannot distinguish a precise directionality of the growth (N- or C-terminal) for the HET-s prion, it is possible that

additional factors in vivo may determine it. For example, specific proteins capping one of the two ends would exert such an effect.

Overall, the results reported in this work show that refined computational tools like SCPS can provide valuable insight into the atomistic details of misfolding events, that may inspire original hypotheses laying the groundwork for new experimental approaches. The concordance between the results of our computational studies on HET-s and PrP$^{Sc}$ provides the first evidence that prions from distant species may propagate similarly, implying that this common replication mechanism could have been conserved through evolution.

## Methods

### Implementation of Self-Consistent Path Sampling

SCPS is an iterative enhanced path sampling algorithm derived directly from the Langevin equation which can be used to effectively generate ensembles of transition pathways connecting a given ensemble of configurations to configurations in a target state. Its implementation consists of three steps (Fig 1).

*STEP 1: rMD simulations*. Starting from a given initial condition in the reactant, an ensemble of reactive trajectories is generated using the rMD algorithm. In these simulations, an external biasing force is introduced to impair backtracking towards previously visited states defined along a single reaction coordinate $z(X)$. The biasing force acting on a single atom is computed as:

$$\mathbf{F}_i(X, z_m) = -k_R \ \nabla_i z(X)[z(X) - z_m] \cdot \theta[z(X) - z_m(t)], \tag{1}$$

where $\theta(x)$ is the Heaviside step-function, $k_R$ determines the strength of the biasing force, $z(X)$ is the CV which represents our initial guess for the reaction coordinate and $z_m(t)$ is the minimum value attained by $z(X)$ up to time $t$. In the specific case of protein folding simulations and the present simulation of HET-s propagation, we adopted a definition of $z(X)$ based on the overlap between the instantaneous contact map and the target contact map:

$$z(X) = \sum_{|i-j|>35}^{N} [C_{ij}(X) - C_{ij}(X_{\text{Ref}})]^2 \tag{2}$$

In this equation, $i$ and $j$ label the atoms and $X_{\text{Ref}}$ is a reference three-dimensional structure in the target state. The constraint $|i\text{-}j| > 35$ is introduced to exclude the trivial contacts due to bonded interactions within amino-acids. The entries of the contact map are a continuous function of atomic configuration, defined as:

$$C_{ij}(X) = \frac{1 - \left( |\mathbf{x}_i - \mathbf{x}_j| / r_0 \right)^6}{1 - \left( |\mathbf{x}_i - \mathbf{x}_j| / r_0 \right)^{10}} \tag{3}$$

where $r_0$ is a reference distance, set to 7.5 Å, while $\mathbf{x}_i$ and $\mathbf{x}_j$ are the atomic coordinates.

We note that, according to Eq 1 the external force is introduced only when the system backtracks toward the reactant state (i.e. when $z(X) > z_m$). Otherwise, the bias remains latent and the system spontaneously proceeds as in plain MD. At the end of the simulation, a trajectory is considered to have successfully reached the target state if a frame $n$ satisfies the following

conditions:

$$RMSD(n) < RMSD_{th} \tag{4}$$

$$\sqrt{\frac{1}{(N_F - n + 1)} \sum_{k=n}^{N_F} \theta[RMSD(k) - RMSD_{th}] \cdot [RMSD(k) - RMSD_{th}]^2} < e_{th} \tag{5}$$

The RMSD to the target reference state is computed for each frame and $RMSD_{th}$ represents the closeness threshold, which was set to 3 Å. In Eq 5, $N_F$ is the total number of trajectory frames and $e_{th}$ is a tolerance threshold. Therefore, to properly consider a trajectory reaching the target reference state, one of its frames must reach below the closeness threshold. The root-mean-squared error, computed on the subsequent frames that are above the threshold, must lay below the tolerance $e_{th}$, that was set to 0.3 Å.

*STEP 2*: *Calculation of the Mean Transition Paths*. The contact map entries $C_{ij}(X)$ calculated for the frames of all the rMD trajectories reaching the target state starting from an identical initial conformation are averaged to estimate their iso-time mean, resulting in the mean-path of the folding process:

$$\langle C_{ij}(t) \rangle = \frac{1}{N_T} \sum_{n=1}^{N_T} C_{ij}^n[X(t)] \tag{6}$$

Here, $C_{ij}^n[X(t)]$ is an element of the contact map computed at time $t$ for the $n_{th}$ trajectory, and $N_T$ is the total number of successful reactive trajectories generated from that initial condition. For convenience, the mean path is downsampled to a low number of contact maps equally spaced in $z(X)$ distance, namely $\langle C \rangle_{k=1 \dots NC}$.

The mean path is then used to define two new coordinates:

$$s_\lambda(X) = 1 - \frac{\frac{1}{N_C} \sum_k^{N_C} k e^{-\lambda \|C(X) - \langle C \rangle_k\|^2}}{\sum_k^{N_C} e^{-\lambda \|C(X) - \langle C \rangle_k\|^2}} \tag{7}$$

$$w_\lambda(X) = \frac{1}{\lambda} \ln \sum_k^{N_C} k e^{-\lambda \|C[X] - \langle C \rangle_k\|^2} \tag{8}$$

where $\|\dots\|$ denotes the norm defined in Eq 2, $\langle C \rangle_k$ is the $k_{th}$ contact map along the mean path, $N_C$ is the total number of contact maps. In the large $\lambda$ limit, the former collective variable represents the progress of the reaction taking the mean path as reference. By definition $s_\lambda(t)$ is 1 in the unfolded state and 0 in the native state, while the second coordinate $w_\lambda(X)$ measures the shortest distance of the configuration $X$ from points in the mean path. In practice, to ensure computational efficiency and numerical stability, the $\lambda$ parameter is set equal to the average distance between neighboring contact maps along the mean-path.

*STEP 3*: *Self-Consistent Refinement of the RC*. A new set of folding trajectories is generated by employing a modified version of the rMD algorithm, introducing two biasing forces analog to the one defined in Eq 1, but using $s_\lambda(X)$, and $w_\lambda(X)$ instead of $z(X)$ as collective variables, respectively. The productive trajectories generated this way (as defined in Eqs 4 and 5) are then used to compute a new mean path. In turn, an updated version of the $s_\lambda(X)$, and $w_\lambda(X)$ collective variables is computed.

Steps 2 and 3 are iterated until convergence, that is when a new iteration produces identical results to the previous one, according to some arbitrary convergence criterion.

## SCPS simulations of fast-folding proteins

The PDB codes of the 5 fast-folding proteins we considered are 2JOF, 2F4K, 2F21, 2HBA and 2P6J. The initial unfolded conformations were retrieved from the plain MD trajectories by sampling states with fraction of native contacts (Q) lower than 0.1 that are separated by at least one folding-unfolding event. Proteins topologies were generated using Charmm22* force field with TIPS3P water model. Lys, Arg, Asp and Glu residues as well as the N-termini and C-termini were treated in their charged states. His residues were neutral in all proteins except in Villin headpiece, where the single His was protonated. Each initial condition was solvated in a cubic box with edge 20 Å greater than the maximum diameter of the system and neutralized with the appropriate number of $Na^+$ and $Cl^-$ ions. Each system was then energy minimized using the steepest descent algorithm then equilibrated in a 500 ps NVT simulation carried out by restraining the protein heavy atoms using a harmonic potential with constant 1000 kJ $mol^{-1}$ $nm^{-2}$.

For each initial conformation, 20 rMD simulations were performed, consisting in $1.5 \cdot 10^6$ steps employing the leap-frog integrator with 2 fs time-step. In these simulations, the long-range coulombic interactions were treated with Smooth Particle-Mesh Ewald (SPME) with a cutoff of 9 Å for the short-range interactions using Verlet lists. Cutoff for Van der Waal interaction was set to 9 Å. Bonds with hydrogen atoms were constrained using LINCS [28]. We used the Nose-Hoover thermostat [29] with a time constant of 1 ps. The values of $k_R$ are reported in S1 Table while the contact distance $r_0$ was set to 7.5 Å. A cutoff radius, $r_c$, of 12 Å was also introduced to increase the computational efficiency, so that for all $r_{ij} > r_c$ the contact map entries are set to 0. For each condition, the folding trajectories (defined according to Eqs 4 and 5) were used to compute the mean path in the space of the contact maps, as described in Eq 6. Each mean path was downsampled to 10 contact maps, equally spaced in the $z(X)$ distance. The mean paths were used to build the CVs used in the subsequent SCPS iteration. The values of $k_s$ and $k_w$ (the ratchet constants of $s_\lambda$ and $w_\lambda$ respectively) for each condition are reported in S1 Table. We performed a total of three SCPS iterations for each protein.

## Comparison of SCPS generated folding pathways with plain MD trajectories

Folding events sampled by rMD and SCPS were compared with the reactive portion of the plain MD trajectories (folding and time-reversed unfolding), defined as the interval of frames starting from a low threshold, fraction of native contacts (Q) < 0.1, and arriving to a high threshold, Q > 0.9. To assess the agreement of rMD and SCPS trajectories with plain MD data, we adopted the path similarity definition introduced in Ref. [14]:

$$s(k, k') = \frac{2}{N(N-1)} \sum_{i<j} \delta[M_{ij}(k) - M_{ij}(k')] \tag{9}$$

where $k$ and $k'$ refer to two different trajectories and $M_{ij}(k)$ is an element of a matrix M which describes the relative order of contacts formation for trajectory $k$ and it is defined as:

$$M_{ij}(k) = \begin{cases} 1 & t_i(k) < t_j(k) \\ 0 & t_i(k) > t_j(k) \\ 1/2 & t_i(k) = t_j(k) \end{cases} \tag{10}$$

with $i$ and $j$ running over all native contacts between $C\alpha$ atoms ($i < j$ to remove duplicates) and $t_i(k)$, $t_j(k)$ the times at which they are formed. A contact was considered to be formed after at least 5 consecutive frames in which the distance between the two $C\alpha$ atoms was less or equal

than 7.5 Å. This similarity definition implies that $s(k, k') = 1$ if all the native contacts in $k$ and $k'$ are formed in the same order, while it is 0 if they are formed in a completely different order.

A distribution of self-similarity, $A$, was computed using Eq 9 for all possible pairs of plain MD reactive trajectories. Such distribution highlights the intrinsic degree of heterogeneity of the folding pathways computed by plain MD. Distributions of cross-similarity, $R_i$, were then computed for all the proteins at each iteration step, between pairs in which the $k$ trajectory is selected from plain MD and the $k'$ from the biased folding events. A random reference distribution was generated to compare the similarity of rMD and SCPS to plain MD with the similarity between plain MD and a random sequence of native contact formation. To this end, an ensemble of $10^5$ random cross-similarity distributions were generated for each protein. Each random cross-similarity distribution is computed between the order of contact formation matrices coming from the Anton trajectories and an equal number of random matrices generated as in Eq 10 with series of contacts formation times randomly extracted with equal probability. The degree of agreement between each cross-similarity $R_i$ distribution and $A$ was computed by means of the Kullback-Leibler divergence, $D_{KL}$, defined as:

$$D_{KL}(A \parallel R_i) = \int_0^1 a(x) \log_2\left(\frac{a(x)}{r_i(x)}\right) dx \tag{11}$$

## Simulations of HET-s propagation

SCPS simulations of HET-s propagation were performed using the Charmm36m force field and the modified Charmm TIP3P water model [30]. Initial conformations, each one consisting of a HET-s dimer in the amyloid form and an unstructured monomer, were generated by thermal denaturation. In particular, the trimeric amyloid structure (PDB 2KJ3) was positioned in a cubic box with an approximate side length greater than 120 Å that was filled with TIP3P water molecules. The system was neutralized with 3 Cl⁻ ions and energy minimized with the steepest descent method. Then, 200 ps of NVT equilibration using the V-rescale thermostat at 800 K were carried out by employing position restraints with a force constant of $10^3$ kJ·mol$^{-1}$·nm$^{-2}$ on heavy atoms. Finally, 5 simulations were performed for 2 ns each at 800 K by releasing the restraints on the N-terminal monomer and the other 5 by releasing the restraints on the C-terminal monomer (S2 Fig). The sampled initial conformations were re-solvated in a smaller cubic box with an approximate side of 100 Å; then 3 Cl⁻ ions were added to counterbalance the protein charge and the system was brought to a final 150 mM NaCl concentration. After energy minimization, 500 ps of MD in the NVT ensemble (at temperature 310 K) followed by 500 ps of MD in the NPT ensemble (with temperature set 310 K, and pressure to 1 Bar) were carried out for each condition with position restraints on the heavy atoms. The V-rescale thermostat and the Parrinello-Rahman barostat were employed for equilibrations and the subsequent rMD and SCPS production. First, 20 rMD simulations for each initial condition were performed, consisting of $3 \cdot 10^6$ steps employing the leap-frog integrator with 2 fs time-step. Frames were saved every 500 steps. The value of $k_R$ was set to $5 \cdot 10^{-5}$ kJ mol$^{-1}$ and $r_0$ to 7.5 Å. The cutoff radius to ignore contact pairs was 12 Å.

The trajectories successfully reaching the target state according to Eqs 4 and 5 were used to compute the mean path of each condition (Eq 6), which was down-sampled to 10 equally spaced contact maps. From each initial condition, 20 simulations were then re-launched for SCPS with bias constants $k_s = 1.5 \cdot 10^{-5}$ kJ mol$^{-1}$ and $k_w = 3 \cdot 10^{-5}$ kJ mol$^{-1}$. A total of 2 SCPS iterations for each initial condition were performed. In this work, we adopted the convergence criterion based on the structure of reaction pathways in the plane selected by the RMSD to the native structure of the two terminal rungs (see Fig 4).

### Analysis of HET-s simulations

The analysis regarding the order of rung formation was first performed by computing the RMSD to the target state of the rungs of the converting monomer. The all-atom RMSD of the two rungs (N-term: R225 to V245, C-term: T261 to Y281, rungs depicted in Fig 1) was evaluated for all the trajectories successfully reaching the target state. The calculation of the RMSD was performed with alignment on the two monomers in the amyloid state (the converting monomer was not included for the alignment).

For each iteration, two different probability densities were generated using frequency histograms: the first was constructed using all the productive trajectories in which the HET-s propagation starts at the N-terminus of the fibril. The second was obtained collecting all the productive paths in which the HET-s propagation starts at the C-terminus of the fibril. It is important to emphasize that the negative logarithms of these densities are not directly related to a free energy distribution, because SCPS reactive trajectories are intrinsically out of equilibrium.

We computed the median point in the reaction progress at which each residue of the converting monomer assumes a β-strand structure. A residue was considered to form a β-strand the first time that such conformation was kept for more than 5 frames consecutively. The time of formation was then converted to the corresponding value of Q (corresponding to the fraction of reference contacts). Calculations of the secondary structures were performed with the STRIDE algorithm [31]. Finally, PCA analysis was performed by using the Cα contact maps and the all-atom contact maps (excluding hydrogens). For the N-terminal propagation simulations, the contact maps were calculated using all the residues of the converting monomer and the residues belonging to the N-terminal rung of the N-terminal monomer already included in the fibril in the initial state. For the C-terminal propagation simulations, residues of the converting monomer and the C-terminal rung of the C-terminal monomer already included in the fibril were used for contact map calculation. In this all-atom PCA analysis, each trajectory was down-sampled to a total of 300 frames.

### Data production, analysis, and visualization software

Biased simulations (rMD and SCPS) were performed in Gromacs 2018 [32], where we implemented the collective variables $z(X)$, $s_\lambda(t)$ and $w_\lambda(t)$. Data analysis was performed in python using the following libraries: MDAnalysis, NumPy and SciPy. Python scripts were accelerated with the Numba compiler. Graphs were obtained by using Matplotlib in python. Images of protein conformations were generated in UCSF Chimera.

### Supporting information

**S1 Fig. Path similarity distributions of the fast-folding proteins.** In each graph, three path similarity distributions are represented: (i) the path similarity distribution obtained by comparing the order of native contact formation between biased folding trajectories (rMD or SCPS) with the plain-MD folding trajectory (defined as cross-similarity $R_i$, blue; where $i$ is the iteration number); (ii) the path similarity distribution computed by comparing the order of native contact formation between the folding plain-MD trajectories within themselves (defined as self-similarity, $A$, grey); (iii) the path similarity obtained by comparing the plain-MD trajectories with random sequences of native contact formation (defined as random, $R_r$, dashed line).
(TIF)

**S2 Fig. Initial conditions of HET-s simulations.** The initial conditions used to generate the propagation pathways are reported. Initial conditions for simulating propagation from the fibril N-terminus were obtained by performing high-temperature MD, introducing positional restraints on heavy atoms on the two C-terminal monomers. Initial conditions for simulating propagation from the fibril C-terminus were obtained by performing high-temperature MD, introducing positional restraints on heavy atoms on the two N-terminal monomers.
(TIF)

**S3 Fig. PCA of the HET-s propagation trajectories using all no-H atoms.** Graphs on the left represent the free energy landscape in the principal component plane of the trajectories propagating from the fibril N-terminus (**A**) and C-terminus (**B**), respectively. Bar plots on the right show the contribution of the contact-type sets for the two principal components.
(TIF)

**S1 Table. Additional information regarding the simulations of the fast-folding proteins.** In this table, additional information regarding the folding simulations are reported. $N_C$ is the number of initial conditions, $T$ is the simulations temperature, $k_R$ is the ratchet force constant, $k_s$ and $k_w$ are the SCPS force constants, $N_{FC}$ is the number of sets (each set start from a different initial condition) for which at least one folding event is observed and $< NF >$ is the average number of folding trajectories for each set.
(TIF)

**S1 Movie. Propagation of the Het-s prion from the N-terminal surface of the fibril.** Visualization of a representative SCPS trajectory of a Het-s monomer converting at the N-terminal surface of the fibril.
(MOV)

**S2 Movie. Propagation of the Het-s prion from the C-terminal surface of the fibril.** Visualization of a representative SCPS trajectory of a Het-s monomer converting at the C-terminal surface of the fibril.
(MOV)

## Acknowledgments

We thank DE Shaw Resarch for making available the results of their protein folding simulation.

## Author Contributions

**Conceptualization:** Giovanni Spagnolli, Jesús R. Requena, Emiliano Biasini, Pietro Faccioli.

**Data curation:** Luca Terruzzi, Giovanni Spagnolli, Alberto Boldrini.

**Formal analysis:** Luca Terruzzi, Giovanni Spagnolli, Alberto Boldrini, Pietro Faccioli.

**Funding acquisition:** Emiliano Biasini, Pietro Faccioli.

**Investigation:** Luca Terruzzi, Giovanni Spagnolli, Alberto Boldrini, Emiliano Biasini, Pietro Faccioli.

**Methodology:** Luca Terruzzi, Giovanni Spagnolli, Alberto Boldrini, Emiliano Biasini, Pietro Faccioli.

**Project administration:** Emiliano Biasini, Pietro Faccioli.

**Software:** Luca Terruzzi, Giovanni Spagnolli, Alberto Boldrini.

**Supervision:** Giovanni Spagnolli, Emiliano Biasini, Pietro Faccioli.

**Validation:** Luca Terruzzi, Alberto Boldrini.

**Visualization:** Luca Terruzzi, Giovanni Spagnolli, Alberto Boldrini.

**Writing – original draft:** Giovanni Spagnolli, Emiliano Biasini, Pietro Faccioli.

**Writing – review & editing:** Luca Terruzzi, Alberto Boldrini, Jesús R. Requena, Pietro Faccioli.

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
