## [Decision Letter · Decision Letter 0]

15 Jun 2020

Dear Prof. faccioli,

Thank you very much for submitting your manuscript "All-Atom Simulation of the HET-s Prion Replication" for consideration at PLOS Computational Biology.

As with all papers reviewed by the journal, your manuscript was reviewed by members of the editorial board and by several independent reviewers. In light of the reviews (below this email), we would like to invite the resubmission of a significantly-revised version that takes into account the reviewers' comments.

We cannot make any decision about publication until we have seen the revised manuscript and your response to the reviewers' comments. Your revised manuscript is also likely to be sent to reviewers for further evaluation.

Sincerely,

Eugene I. Shakhnovich

Guest Editor

PLOS Computational Biology

Nir Ben-Tal

Deputy Editor

PLOS Computational Biology

Reviewer's Responses to Questions

**Comments to the Authors:**

Reviewer #1: Please see attached file

**Have all data underlying the figures and results presented in the manuscript been provided?**

Reviewer #1: None

PLOS authors have the option to publish the peer review history of their article (what does this mean?). If published, this will include your full peer review and any attached files.

Reviewer #1: No
---

## [Editor Report · Decision Letter 1]

12 Jul 2020

Dear Prof. faccioli,

Thank you very much for submitting your manuscript "All-Atom Simulation of the HET-s Prion Replication" for consideration at PLOS Computational Biology.

As with all papers reviewed by the journal, your manuscript was reviewed by members of the editorial board and by several independent reviewers. In light of the reviews (below this email), we would like to invite the resubmission of a significantly-revised version that takes into account the reviewers' comments.

In order to evaluate this revision we need a version where changes introduced to address referee's critiques are clearly marked in the text of revised paper (e.g. in red font or in other easily identifiable way). Please prepare a version with changes marked and resubmit.

We cannot make any decision about publication until we have seen the revised manuscript and your response to the reviewers' comments. Your revised manuscript is also likely to be sent to reviewers for further evaluation.

Sincerely,

Eugene I. Shakhnovich

Guest Editor

PLOS Computational Biology

Nir Ben-Tal

Deputy Editor

PLOS Computational Biology

In order to evaluate this revision we need a version where changes introduced to address referee's critiques are clearly marked in the text of revised paper (e.g. in red font or in other easily identifiable way). Please prepare a version with changes marked and resubmit.
---

## [Editor Report · Decision Letter 2]

28 Jul 2020

Dear Prof. faccioli,

Thank you very much for submitting your manuscript "All-Atom Simulation of the HET-s Prion Replication" for consideration at PLOS Computational Biology. As with all papers reviewed by the journal, your manuscript was reviewed by members of the editorial board and by several independent reviewers. The reviewers appreciated the attention to an important topic. Based on the reviews, we are likely to accept this manuscript for publication, providing that you modify the manuscript according to the review recommendations.

Please provide a clean version with all changes accepted  with significant ones (.1 sentence) marked in (special font color, italic, bold whatever you prefer). Track changes tool makes current copy unreadable especially  when all comments are kept in Italian!

Sincerely,

Eugene I. Shakhnovich

Guest Editor

PLOS Computational Biology

Nir Ben-Tal

Deputy Editor

PLOS Computational Biology

[LINK]
---

## [Editor Report · Decision Letter 3]

3 Aug 2020

Dear Prof. faccioli,

We are pleased to inform you that your manuscript 'All-Atom Simulation of the HET-s Prion Replication' has been provisionally accepted for publication in PLOS Computational Biology.

Best regards,

Eugene I. Shakhnovich

Guest Editor

PLOS Computational Biology

Nir Ben-Tal

Deputy Editor

PLOS Computational Biology

---

## [Editor Report · Acceptance letter]

11 Sep 2020

PCOMPBIOL-D-20-00698R3

All-atom simulation of the HET-s prion replication

Dear Dr faccioli,

I am pleased to inform you that your manuscript has been formally accepted for publication in PLOS Computational Biology. Your manuscript is now with our production department and you will be notified of the publication date in due course.

With kind regards,

Laura Mallard
